# The Role of the Gastrointestinal Microbiome in Liver Disease

**DOI:** 10.3390/pathogens12091087

**Published:** 2023-08-27

**Authors:** Nicholas Shalaby, Dorit Samocha-Bonet, Nadeem O. Kaakoush, Mark Danta

**Affiliations:** 1School of Clinical Medicine, Faculty of Medicine and Health, University of New South Wales, St Vincent’s Healthcare Campus, Darlinghurst, NSW 2010, Australia; 2Clinical Insulin Resistance Group, Garvan Institute of Medical Research, Darlinghurst, NSW 2010, Australia; 3School of Biomedical Sciences, Faculty of Medicine and Health, University of New South Wales, Kensington, NSW 2033, Australia; 4Department of Gastroenterology and Hepatology, St Vincent’s Hospital, Darlinghurst, NSW 2010, Australia

**Keywords:** microbiome, liver fibrosis, cirrhosis, gastrointestinal, antifibrogenic

## Abstract

Liver disease is a major global health problem leading to approximately two million deaths a year. This is the consequence of a number of aetiologies, including alcohol-related, metabolic-related, viral infection, cholestatic and immune disease, leading to fibrosis and, eventually, cirrhosis. No specific registered antifibrotic therapies exist to reverse liver injury, so current treatment aims at managing the underlying factors to mitigate the development of liver disease. There are bidirectional feedback loops between the liver and the rest of the gastrointestinal tract via the portal venous and biliary systems, which are mediated by microbial metabolites, specifically short-chain fatty acids (SCFAs) and secondary bile acids. The interaction between the liver and the gastrointestinal microbiome has the potential to provide a novel therapeutic modality to mitigate the progression of liver disease and its complications. This review will outline our understanding of hepatic fibrosis, liver disease, and its connection to the microbiome, which may identify potential therapeutic targets or strategies to mitigate liver disease.

## 1. Introduction

Liver disease causes approximately two million deaths a year, posing a significant global health problem [1]. This is the consequence of several aetiologies, including alcohol-related and metabolic-related disease, viral infections, cholestasis and immune disease, leading to fibrosis and, eventually, cirrhosis. While various pharmacological agents are under study, there are currently no specific antifibrotic therapies available to reverse liver fibrosis, so treatment aims at managing the underlying factors to mitigate the development of liver disease [2]. Given the bidirectional relationship between the liver and gastrointestinal tract (GIT) mediated through the gastrointestinal microbiota, there is increasing interest in the gastrointestinal microbiome and its capacity to affect liver disease and how this may be manipulated therapeutically.

## 2. Liver Fibrosis

Liver fibrosis is the response to chronic inflammation that results in increased extracellular deposition of fibres, including collagen I, III and IV, fibronectin, undulin, elastin, laminin, hyaluronan and proteoglycans, and reduced breakdown of the hepatic extracellular matrix (ECM) [3]. It is primarily mediated by activated hepatic stellate cells (HSCs) [4]. Typically, HSCs are found within the Space of Disse of the hepatic sinusoids, maintaining the homeostasis of the hepatic environment and ECM by regulating hepatic stem cells and hepatic fibrosis. When activated, HSCs become fibroblast-like cells which are responsible for the increased production of collagen in the ECM. The development of progressive fibrosis and, eventually, cirrhosis leads to hepatocellular dysfunction, the development of portal hypertension and the risk of dysplasia and malignancy. Hepatic failure, termed decompensation, develops in up to 11% of individuals with cirrhosis annually [5]. Complications of cirrhosis include infection, ascites and oedema, variceal bleeding, spontaneous bacterial peritonitis, hepatic encephalopathy, and hepatocellular carcinoma. It is now recognised that removal or control of the underlying factors can lead to improvement in hepatic fibrosis. Activated HSCs can be deactivated through three pathways: HSCs undergoing apoptosis, becoming senescent, or reverting to an inactivated form [6]. Interestingly, changes to the diversity, composition and function of the gastrointestinal microbiota, in conjunction with its bidirectional interactions with the liver, may contribute to hepatic fibrosis. This review will outline our understanding of hepatic fibrosis and its connection to the microbiome, highlighting potential therapeutic targets or strategies to mitigate liver disease.

### 2.1. Common Pathophysiological Pathways of Liver Fibrosis

There are several common and disease-specific pathways associated with liver fibrosis. Damage to hepatocytes is the typical inciting fibrogenic event, although some aetiologies, such as HIV/HCV coinfection can activate HSCs independent of hepatocyte damage [7]. This results in the release of reactive oxygen species (ROS) and fibrogenic mediators, hepatocyte apoptosis and the replacement of the typical endothelial fenestrations of the hepatic sinusoids with fibronectin and collagen I and III, termed sinusoid capillarisation [8]. Following this, there is the activation of inflammatory cells, including Kupffer cells, which release inflammatory cytokines and more ROS [9]. This inflammatory cascade leads to the recruitment of white blood cells, resulting in the downstream migration and activation of HSCs. When HSCs are activated, they are changed from a star-shaped cell to an elongated myofibroblast which further perpetuates the inflammatory cycle through the release of cytokines, expression of cell adhesion molecules as well as activation and modulation of lymphocytes (Figure 1).

One of the most potent mediators of this inflammatory cascade is transforming growth factor-beta (TGF-beta). TGF-beta is a cytokine released by HSCs, Kupffer cells and hepatocytes in a latent form. This inactive form is bound to the ECM via latent TGF-beta binding protein for subsequent activation by proteases. While TGF-beta is primarily involved in the TGF-beta/suppressor of mothers against decapentaplegic (SMAD) pathway, there is significant crosstalk with other pathways, including the classical mitogen-activated protein kinase (MAPK) and Janus kinase-signal transduction and activators of transcription (Jak-STAT) pathways. The activation of TGF-beta occurs with sinusoid capillarisation and interaction with fibrotic proteins, including collagen IV, fibrinogen and urokinase-type plasminogen activator [10]. TGF-beta expands the inflammatory response further by activating HSCs and inhibiting matrix metalloproteinases, which are responsible for breaking down the ECM. Additionally, TGF-beta induces greater fibrosis by increasing the effects of other molecules. Platelet-derived growth factor (PDGF) and epidermal-derived growth factor (EGF) are two such molecules acting as potent attractors and proliferators of HSCs following release from platelet alpha granules and activated macrophages during inflammation [11]. TGF-beta also appears to interact with the sympathetic nervous system. This may occur via the upregulation of TGF-beta by noradrenaline produced by dopamine beta dehydroxylase found in HSCs [12].

Another significant fibrotic pathway is the renin−angiotensin system (RAS). The liver has a local RAS with dramatically increased expression in activated HSCs. Systemic renin converts angiotensinogen, stored in hepatocytes, to angiotensin I. This is subsequently converted to angiotensin II by hepatic angiotensin-converting enzyme (ACE). Interestingly, the exact cell that releases hepatic ACE has not been identified. Angiotensin II is a potent vasoconstrictor released by HSCs only once they have been activated. Angiotensin II binds to the angiotensin II receptor type 1 (AT1) of mast cells resulting in the release of TGF-beta, further increasing liver fibrosis [13]. Its primary actions in liver fibrosis are recruiting inflammatory cells and inducing the expression and secretion of ECM proteins while inhibiting collagen degradation and acting as a mitogen for activated HSCs [14,15]. Further, angiotensin II release is increased by PGDF and its synthesis is stimulated by EGF, Thrombin and Endothelin-1 [16]. Endothelin-1 is produced by HSCs, and its release is stimulated by angiotensin II, creating positive feedback. It acts on the Endothelin-A receptors on HSCs causing activation in a paracrine and autocrine manner. The other less strongly activated receptor, Endothelin-B, found on sinusoidal endothelium and HSCs, triggers the release of nitric oxide, which results in HSC relaxation [17]. Nitric oxide, as well as prostaglandin E2 and adrenomedullin, oppose angiotensin II secretion and hence are antifibrogenic.

Similar to angiotensin II, leptin is expressed by activated but not quiescent HSCs. Leptin is an adipokine released by HSCs in the liver and by adipocytes systemically. Its primary targets include sinusoidal endothelial cells and Kupffer cells. Leptin has been shown to increase fibrosis and procollagen type I expression in carbon tetrachloride rat models [18]. It appears that leptin results in increased TGF-beta expression in the liver [19]. However, as leptin induces a significant amount of fibrosis, it is unlikely that TGF-beta release is the only vector by which leptin increases fibrosis. Leptin may also stimulate inflammation independent of TGF-beta by increasing inflammatory factors such as tumour necrosis factor-alpha (TNF-α) [18].

Adiponectin, another adipokine produced systemically by adipocytes and locally by activated HSCs, has an antifibrotic effect on the liver. This is due to the suppression of PDGF, which reduces the proliferation and migration of HSCs as well as the positive feedback loop TGF-beta 1 has on TGF-beta 1 gene expression, connective tissue growth factor and nuclear translocation of Smad2 [20].

Other mediators including monocyte chemotactic protein type-1 (MCP-1), which attracts and proliferates HSCs, macrophage inflammatory protein-2 (MIP-2), interleukin-8 (IL-8) and Regulated upon Activation, Normal T Cell Expressed and Presumably Secreted (RANTES), are also involved in fibrosis. RANTES increases intracellular calcium concentration and free radical formation from HSCs while also inducing HSC proliferation and migration [21,22]. RANTES is also a powerful attractor of monocytes, eosinophils and activated CD4 T-cells, further perpetuating the inflammatory cascade [22].

### 2.2. Liver Fibrosis in Alcohol-Related (ALD) and Metabolic Dysfunction-Associated Steatotic Liver Disease (MASLD)

The American Association for the Study of Liver Disease (AASLD) has recently changed the nomenclature of nonalcoholic fatty liver disease (NAFLD). Steatotic liver disease (SLD) is the overarching condition. In the context of metabolic syndrome, this is now termed metabolic dysfunction-associated steatotic liver disease (MASLD) instead of NAFLD. Metabolic dysfunction-associated steatohepatitis (MASH) replaces non-alcoholic steatohepatitis (NASH). When a combination of alcohol and metabolic dysfunction contributes, this is termed MetALD. Finally, when metabolic syndrome is not present, then SLD should be used [23].

Different aetiologies can have specific fibrogenic mechanisms and fibrosis patterns. ALD and MASLD characteristically have a “chicken wire” pattern of pericellular fibrosis around groups of steatotic hepatocytes in lobular zone 3, as well as capillarisation of sinusoids [24]. This leads to septa formation isolating regenerating nodules. In ALD, the metabolisation of ethanol to acetaldehyde activates HSCs, increases gene transcription and extracellular matrix synthesis in activated HSCs and releases inflammatory mediators. Further, ethanol exposure can lead to lipid peroxidation, and the products of this reaction can increase HSC activation [25,26].

The pathogenesis of liver fibrosis in MASLD and MASH is postulated to be due to a multiple parallel hits model [27]. These multiple hits may result from an accumulation of lipids in the liver, long term/chronic consumption of synthetic industrially-added fructose, trans fat and aryl hydrocarbon receptor ligands, the effects of endotoxin from the microbiota and its mediators, the action of adipocytokines such as adiponectin and leptin and effects of the innate immune system, which together cause repeated inflammatory insults and steatosis in the liver. This induces hepatocyte apoptosis and leads to the recruitment of inflammatory cells, ultimately resulting in fibrosis [28].

### 2.3. Liver Fibrosis in Viral and Autoimmune Hepatitis

Chronic viral hepatitis (such as caused by hepatitis B, C and Delta) and autoimmune hepatitis develop portal-central venule bridging necrosis, which leads to portal-central septa formation. This is usually associated with interface hepatitis, with cell-mediated immune inflammation at the interface of parenchyma and connective tissue of the portal zone. These individuals are also predisposed to earlier development of portal hypertension as a result of the disruption to the vascular connections within the portal circulation [29,30,31]. Viruses escape the surveillance of the human leukocyte antigen-II (HLA-II) directed immune response leading to infection of hepatocytes [32]. The cell-mediated injury leads to the production of fibrogenic viral proteins, oxidative stress as well as inflammatory cell recruitment, which increases the activation of HSCs.

### 2.4. Liver Fibrosis in Cholestatic Disease

Cholestatic disease, including primary biliary cholangitis (PBC) and primary sclerosing cholangitis (PSC), both result in biliary fibrosis. This type of fibrosis develops in a portal-to-portal pattern as a result of the proliferation of both reactive bile ductules and periductal fibroblast-like cells at the portal−parenchymal interface. This progresses to portal−portal septa surrounding nodules of regenerating parenchyma. The connections between the central venules and portal triads are preserved until the late stages of fibrosis [33,34]. Cholestatic liver fibrosis occurs due to the release of TGF-beta and activation of T-cells resulting in the injury of the bile ducts. This damage induces the secretion of fibrogenic mediators by biliary cells, which causes portal myofibroblasts and perisinusoidal HSCs to become activated and release extracellular matrix [35].

## 3. Microbiome and the Liver

The gastrointestinal microbiome refers to the dynamic and individualised ecosystem of bacteria, fungi, protozoa, and viruses that reside along the gastrointestinal tract as well as their genomic content (metagenomes). They are important contributors to both physiological and pathological processes within the body [36,37]. There is a symbiotic relationship between the host and its microbiota, whereby the gastrointestinal microbiota supports immune responses, epithelial functional integrity and the metabolism of nutrients and nonnutritive food ingredients that would otherwise be unabsorbable [38].

The development of the microbiome, which begins after birth and throughout the first few years of life, is strongly influenced by breastfeeding and environmental exposure [39]. Following this, the gut microbiota remains relatively stable in most individuals, although it can be altered by changes to the diet, antibiotic use and during periods of illness. The gastrointestinal microbiota varies in composition and density throughout the intestine, with 90% of the microbiota comprising four major phyla: Bacteroidetes, Firmicutes, Proteobacteria and Actinobacteria [37]. Recent studies have shown that certain microbial taxa, including *Bifidobacterium, Lactobacillus, Faecalibacterium*, *Roseburia* and *Ruminococcus* spp., are associated with anti-inflammatory effects in the gut as well as benefits to metabolic parameters. Additionally, different species belonging to *Enterobacteriaceae, Enterococcaceae, Clostridiaceae* and *Bacteroidaceae* have been shown to be associated with both pathogenic and beneficial effects on gut health [40,41,42,43,44]. The balance of all the organisms that comprise the microbiota is vital for normal function. Dysbiosis is the disruption to the diversity or composition of the gut microbiota, specifically the loss of symbionts and expansion of pathobionts. While many studies have found a correlation between changes in relative abundance with disease, the mechanistic understanding of this interaction remains to be fully elucidated. Furthermore, identifying and characterising the nature of nonbacterial microbes remains a challenge unmet by current techniques [45].

There are bidirectional feedback loops between the liver and GIT via the portal venous and biliary systems, which are mediated by microbial metabolites, specifically SCFAs and secondary bile acids (Figure 2) [46,47,48,49]. While the GIT and liver have complex connections, the intestinal barrier is important to prevent the translocation of bacteria or endotoxins into the portal circulation. Disruption of these feedback loops and the barriers separating the GIT and liver contributes to liver disease [50].

### 3.1. Bile Acid Conversion

Bile acids have an important role in maintaining gastrointestinal integrity [48,49,51]. Primary bile acids are produced by the liver from cholesterol and released into the GIT where they are then largely reabsorbed in the ileum via enterohepatic circulation. This process is highly regulated via factors that provide information about the lumen, mucosa, hepatocytes and local and systemic inflammation for bile acid production to meet intestinal demands [52]. Recently it has been established that bile acids influence the composition of the microbiota through the taurine component while being transformed into secondary bile acids, which include deoxycholic and lithocholic acids [48,49]. Changes in the amount and ratio of bile acids have been observed in cirrhosis and are associated with dysbiosis [53]. Further, in cirrhosis, primary bile acid production was decreased due to dysfunction of hepatocytes. As a result, populations of bacteria that use primary bile acids as substrate and subsequently convert primary bile acids to secondary bile acids become outcompeted by pathogenic bacteria leading to dysbiosis [53]. The reduced concentration of secondary bile acids results in reduced farnesoid X receptor (FXR) stimulation impacting the repair of the epithelial and GIT vascular barriers and control of metabolic syndrome by increasing glycogen production, decreasing lipogenesis and very low-density lipoprotein production, thereby reducing hepatic glucose and fatty acid output [54,55,56].

### 3.2. Short-Chain Fatty Acids

SCFAs, including acetate, propionate, butyrate and valerate, serve as another group of important metabolites. They are produced through the metabolism of prebiotic food ingredients, including dietary fibre, certain peptides, biosurfactants and vitamins. SCFAs help maintain the intestinal barrier by binding to G-protein coupled receptors (GPCRs), including free fatty acid receptors 1 and 2, and GPR109a to provide energy to enterocytes and inhibit immune cells such as T-regulatory cells and macrophages [57,58,59,60,61,62,63,64]. Additionally, these SCFAs may act on GPCRs to induce secretion of glucagon-like peptide-1 (GLP-1), which has implications for glycaemic control and has been associated with improvements in MASLD [63]. This was identified in a small pilot study conducted on six women with insulin resistance as well as in animal models and so must be confirmed in a more comprehensive human study [63,65]. Further, butyrate can induce gluconeogenic gene transcription in enterocytes, while the action of propionate on the GPCRs induces the conversion of propionate to glucose. These mechanisms alter the composition of the microbiota, reduce adiposity and improve glucose control [60,66,67]. Dysbiosis in liver disease can lead to a reduction in SCFA production by bacteria, therefore contributing to inflammation and increased fibrosis [68,69]. Therefore, SCFA production is both affected by and affects liver disease.

## 4. Dysbiosis in Liver Disease

Dysbiosis is mediated by changes to the intestinal environment due to factors such as genetics, environment, diet, medication, alcohol, or disease. Given the interaction between the liver and the GIT, it is unsurprising that liver disease is associated with intestinal dysbiosis (Table 1).

Dysbiosis has been identified in MASLD both in animal models and in humans. Faecal microbiota transplantation (FMT) is the transplant of faecal microbiota derived from carefully selected healthy donors into a patient’s gastrointestinal tract to alter the composition of the microorganisms. FMT studies have demonstrated the impact of the microbiome on metabolic health. In an important animal study using two mice given a high-fat diet, one of the mice developed hyperglycaemia and had a high plasma concentration of pro-inflammatory cytokines, while the other mouse was normoglycaemic and had a lower systemic inflammation. The microbiota of these mice were then used to colonise germ-free mice which were then also fed the same high-fat diet. The germ-free mice exhibited the same glycaemic profile as their donor, with the hyperglycaemic mice also developing hepatic macrovesicular steatosis [70]. A cross-sectional study that included 53 patients with MASLD and 32 healthy participants, using 16S ribosomal RNA (rRNA) gene amplicon sequencing, found a decrease in *Alistipes*, *Prevotella* and *Ruminococcaceae* and an increase in *Escherichia*, *Anaerobacter*, *Lactobacillus* and *Streptococcus* in individuals with MASLD compared to controls. These changes were associated with the disruption of epithelial tight junctions, increased intestinal permeability, irregularly arranged microvilli, increased inflammatory markers (TNF -α, interleukin-6 and interferon-γ) and decreased immune cells (CD4+ and CD8+ T lymphocytes). This study concluded that a higher relative abundance of *Escherichia* and a lower relative abundance of *Ruminococcaceae*, a family comprising SCFA producers, are likely important factors for MASLD [71]. This study and others confirm the importance of dysbiosis in liver disease [72,75,76,77,79,80]. Interestingly, these studies differ in the specific microbial taxa that have been found to be differentially abundant. This may be due to environmental factors contributing to different microbiota compositions in cohorts, population characteristics, sequencing and diagnostic tools as well as the severity of disease in patients. Importantly, 16S rRNA gene amplicon sequencing has been shown to identify approximately 76% of unique bacterial species in the intestinal microbiome. Although this improves on past studies profiling the microbiota using culturomics (24%), it is still clear that this method is unable to provide a complete representation of the microbiota [94]. In a paediatric cohort with MASH, a cross-sectional study (MASH: *n* = 22, Obese: *n* = 25, Control: *n* = 16) found an increase in *Bacteroides* and *Escherichia* and a decrease in Firmicutes and Actinobacteria using 16S rRNA gene amplicon pyrosequencing. Specifically, this paper suggested that the higher relative abundance of *Escherichia* found in the MASH group and not in the group of children living with obesity not complicated by MASH suggests that a higher relative abundance of these alcohol-producing bacteria may be involved in the pathogenesis of MASH and mirrors findings of other smaller trials [73,78]. Specifically, it was suggested that the increase in alcohol-producing bacteria causes the microbiota to produce more alcohol than in a healthy person leading to a constant source of ROS to the liver, which produces inflammation. This was confirmed in a later study with a prospective observational arm (*n* = 146) measuring fasted post-prandial blood ethanol and an intervention arm (MASLD: *n* = 10, Obese without MASLD: *n* = 10) whereby participants were given a selective alcohol dehydrogenase inhibitor before a mixed meal test. In the observational arm of the study, it was found that blood ethanol was higher in MASLD and increased with disease progression to MASH. In the intervention arm of the study, it was found that inhibition of alcohol dehydrogenase led to a 15-fold increase in peripheral blood ethanol concentrations in individuals with MASLD but disappeared after antibiotic treatment [74]. Further, it was found that MASLD in individuals who are lean and those who have obesity correlate with different microbiota signatures. A 2021 cross-sectional observational study conducted in Japan (obese MASLD: *n* = 51, nonobese SLD: *n* = 51, control: *n* = 87) found, with 16S rRNA gene amplicon sequencing, that the microbiota of nonobese patients with SLD correlated with a significant decrease in *Eubacterium* compared to MASLD patients with obesity [81]. An earlier study from 2019 (lean SLD: *n* = 27, obese MASLD: *n* = 49, control: *n* = 192) found a decrease in *Desulfovibrionaceae* in lean SLD when compared to those with MASLD and obesity [82]. However, observational studies do not confirm a causal link; thus, further studies of higher power and studies utilising animal models are required to elucidate the mechanistic nature of this relationship.

In ALD it is known that the ingestion of alcohol directly affects the microbiome. A small study conducted in mice (alcohol-fed: *n* = 8, control: *n* = 8) found that chronic ethanol feeding over 6 weeks lowered the relative abundance of Bacteroidetes and Firmicutes and increased the relative abundance of Gram negative Proteobacteria, including *Alcaligenes*, and Gram positive Actinobacteria, including *Corynebacterium* [83]. This study also concluded that these changes were associated with increased plasma endotoxin, faecal pH and hepatic inflammation and injury. These results mirrored those of an earlier study which found that after 3 weeks of alcohol feeding, mice had an increased relative abundance of Bacteroidetes and Verrucomicrobia compared to controls. They also found that there was downregulation of bactericidal c-type lectins regenerating islet-derived 3 beta (Reg3b) and gamma (Reg3g) in the small intestine leading to bacterial overgrowth and steatohepatitis in the alcohol-fed group [84]. Interestingly, a study that transplanted human microbiota of patients with ALD into germ-free mice found that severe alcoholic hepatitis was associated with increases in *Bacteroides*, *Bilophila*, *Alistipes*, *Butyricimonas* and *Clostridium* cluster XIVa, as well as decreases in *Faecalibacterium* [85]. Strengthening this relationship, a study conducted on 48 people diagnosed with alcohol use disorder with and without ALD and in 18 healthy subjects found that those with ALD had a lower relative abundance of Bacteroidetes and a higher relative abundance of Proteobacteria that was persistent after extended periods of sobriety. This was correlated with high levels of serum endotoxin [87]. Therefore, it is likely that treatments are required to ameliorate these dysbiotic changes hence justifying the need to explore microbiome-focused treatment modalities. The mechanism by which alcohol impacts the microbiome has been shown to be indirect in a study performed in mice. In this study, no evidence of microbial metabolism of ethanol was found; however, microbiota responded to ethanol by activating acetate dissimilation, hence altering the microbiota composition [86]. Additional studies have attempted to identify whether the severity of ALD affects microbiota composition. Smirnova et al., enrolling individuals with alcoholic hepatitis (*n* = 34), alcohol use disorder (*n* = 20), and healthy controls (*n* = 24), found no differences in composition based on disease severity [88]. In contrast, Lang et al. conducted a larger study (alcoholic hepatitis: *n* = 75, alcohol use disorder: *n* = 43, healthy controls: *n* = 14), which showed that severe alcoholic hepatitis had a decreased relative abundance of *Akkermansia* and increased *Veillonella* [89]. These disparate findings may be due to the smaller participant number in the study conducted by Smirnova and colleagues.

Significant disruption to the balance and diversity of the microbiome is associated with cirrhosis. In a longitudinal study of 244 subjects with compensated cirrhosis (*n* = 121), decompensated cirrhosis (*n* = 98), and healthy controls (*n* = 25), multitagged pyrosequencing of stool samples was used to calculate the relative increases and decreases of the abundance of microbial taxa. This data was then used to create a dysbiosis ratio compared to the controls. When each group of patients was analysed, it was shown that the participants with decompensated cirrhosis had more severe dysbiosis than compensated cirrhosis, although both were worse than controls. Specifically, there was a higher relative abundance of *Enterococcaceae*, *Staphylococcaceae*, and *Enterobacteriaceae* and a reduced relative abundance of Clostridiales XIV, *Ruminococcaceae*, *Lachnospiraceae*, *Veillonellaceae*, and *Porphyromonadaceae*. Further, this dysbiosis was correlated with increased serum endotoxin. Following initial analysis, within a six-month period, another sample was analysed, showing that dysbiosis did not get worse in stable cirrhosis but did become more severe if individuals became decompensated. Patients with decompensated cirrhosis and infections had stool samples taken within 48 h of antibiotic initiation and so antibiotic use should not have affected the analysis. Additionally, this study concluded that in patients with cirrhosis, severe dysbiosis, as calculated by the dysbiosis ratio, was associated with death and organ failure within 30 days [90]. These results aligned with an earlier smaller study (cirrhosis: *n* = 36 and controls: *n* = 24) that revealed the Proteobacteria and Fusobacteria phyla had a higher relative abundance. Further, Firmicutes from the *Veillonellaceae* and *Streptococcaceae* families also had a raised relative abundance, while Bacteroidetes and *Lachnospiraceae* had a lower relative abundance [91]. Another study comparing individuals with cirrhosis (*n* = 98) and healthy controls (*n* = 83) found 28 bacterial species that increased and 38 species that decreased in relative abundance in cirrhosis. Interestingly, they identified that 54% of the increased species were of buccal origin, supporting the notion that buccal microbiota likely colonise the intestinal microbiota in patients with cirrhosis [92]. In contrast, another study comparing the microbiome of American and Turkish populations, with or without cirrhosis (Table 1), found that the Turkish cohort had greater microbial diversity than the American participants, and unlike the Americans, this diversity was not compromised in cirrhosis patients [93]. These differences may be due to a variety of factors related to the environment (including the diet), and disease.

## 5. Intestinal Barrier and Dysfunction

There are three main barriers that prevent the translocation of the microbiota to portal circulation: the mucus barrier, the epithelial layer and the gut vascular barrier. The mucus barrier is composed of two layers; the firm inner layer, which is restrictive to microbial colonisation as a result of antimicrobial peptides, microbiota-excluding proteins and the restricted size of the meshes and the outer layer, which provides an attachment point for bacteria, preventing their removal during peristalsis [95,96,97]. The outer layer also serves as a nutrient source for some bacteria, including *Akkermansia muciniphilia* [98]. The epithelial layer prevents translocation to the portal circulation via tight junctions, antimicrobial peptides, and the electrical repulsion effect of the brush border on the microbiota [99]. The final barrier is the gut vascular barrier, comprising endothelial cells and a network of pericytes and enteric glial cells, together with the immune system [100,101]. In liver disease, the intestinal barrier is often compromised, with tight junction proteins becoming deregulated and the stability of this mucus layer being compromised by mucin-degrading bacteria, especially in the context of a low-fibre diet [98]. These changes can lead to hepatic inflammation and endotoxemia.

Alcohol can lead to several effects on the integrity of the intestinal barrier, including dysbiosis and the toxic effects of alcohol and its byproducts on intestinal epithelial cells. A mouse study exploring the mechanisms of dysbiosis-induced intestinal inflammation in chronic alcohol consumption revealed an association between increased intestinal permeability and liver disease. This study found that 8 weeks of alcohol feeding resulted in jejunal inflammation, which was characterised by increased TNF-α-producing monocytes and macrophages in mice which mirrors biopsies from patients with chronic alcohol abuse. This study also identified myosin light chain kinase and TNF-receptor I as key TNF-α targets, which contribute to intestinal barrier dysfunction and liver disease in chronic alcohol consumption [102]. A smaller trial conducted on 11 male and 14 female participants who were given vodka at 30 min intervals for 4 h revealed that acute binge drinking rapidly increased serum endotoxin and 16S rDNA gene levels suggesting that acute alcohol consumption disrupts the intestinal barrier [103].

Intestinal barrier dysfunction is also a key pathogenic mechanism in MASLD. In an animal study, MASH was induced in mice by feeding them a high-fat or methionine-choline-deficient diet (a classic dietary model for studying MASH) for a week or longer. In this study, it was found that intestinal epithelial barrier and gut vascular barrier dysfunction occurred. The disruption to the gut vascular barrier was found to be dependent on disruption to the Wnt/β-catenin signalling pathway [104]. Another study conducted on mice with a disruption to the gene which encodes for junctional adhesion molecule A, a protein that has been shown to be reduced in MASLD, led to significantly increased mucosal inflammation, tight junction disruption and intestinal epithelial permeability to bacterial endotoxins, compared with control mice when fed a high-fat diet [105]. Human studies have also found these correlations. One such study, conducted on 35 individuals with MASLD, 27 with untreated coeliac disease and 24 healthy controls, assessed small intestine bacterial overgrowth through glucose breath testing, intestinal permeability via urinary excretion of ^51^Cr-ethylene diamine tetraacetate and the integrity of gut tight junctions through immunohistochemical analysis of zona occludens-1 expression in duodenal biopsies. The study found that patients with MASLD had increased intestinal permeability and a higher prevalence of short intestine bacterial overgrowth compared to controls but not as much as bacterial overgrowth found in untreated coeliac patients [106]. Another human study conducted on 21 patients with severe obesity and MASH and nine patients with severe obesity without liver disease found, using the EndoCab assay, which measures antibodies to the core region of endotoxin, that plasma IgG levels against endotoxin were elevated in patients with MASH compared to the controls indicating increased intestinal permeability in these patients [107].

## 6. Dysbiosis, Intestinal Permeability and Liver Disease

While our understanding of the mechanisms of fibrosis due to dysbiosis and intestinal permeability is incomplete, endotoxins seem to play a significant role. The primary endotoxin of concern is lipopolysaccharide (LPS), which is a membrane component of Gram-negative bacteria. In dysbiosis, because there are more of these LPS-containing bacteria and because of disruption to the intestinal barrier, there is an increased risk of endotoxemia and subsequent liver fibrosis. Once translocated, LPS can create a complex with LPS-binding protein, which binds to Toll-like receptor 4 (TLR4). This activates Kupffer cells causing the activation of nicotinamide adenine dinucleotide phosphate (NADPH) oxidase, which in turn increases the production of ROS [108]. ROS modulates inflammation by releasing inflammatory mediators and activating HSCs and Kupffer cell nuclear factor kappa-light-chain-enhancer of activated B cells (NF-KB) which causes increased production of TNF-α. This induces neutrophil infiltration as well as the production of mitochondrial oxidants, causing hepatocyte apoptosis [109]. This relationship was demonstrated in a bile duct ligation model of liver disease in mice with a normal and mutant TLR4, confirming that TLR4 activation by LPS was responsible for fibrogenesis through the upregulation of chemokine secretion and chemotaxis of Kupffer cells as well as the sensitisation of HSCs to TGF-beta [110]. Other bacterial products that can translocate from the gut to the liver include peptidoglycan, lipoteichoic acid, porin and flagellin. The levels of circulating microbial products have been found to be correlated with liver disease severity.

## 7. Modulating the Microbiome in Liver Disease

Given the role of the microbiome in liver disease, it has become a focus as a potential therapeutic target to mitigate liver disease. In particular, intestinal environment modification, targeting microbiota directly and modulating bacterial metabolites and their pathways may provide specific targets to mitigate liver injury.

### 7.1. Modifying the Intestinal Environment

Prebiotics, as part of dietary interventions, can ameliorate dysbiosis, potentially mediating the complex interactions between the microbiome and liver disease. Optimising the substrates available to the microbiota alters the intestinal microbiome and may be a safe and low-cost method of resolving dysbiosis. Specifically, high-fibre diets are associated with a healthy microbiome.

Dietary fibre intake is associated with a healthy microbiome but is neglected in the prototypical Western diet. Some fibres are generally poorly fermented by microbiota but boost intestinal transit, while other fibres are highly fermentable into SCFAs. These fermentable fibres act as a nutrient source for the microbiota, therefore, influencing bacterial growth [111,112]. A systematic review and meta-analysis of 64 studies showed that a high-fibre diet reduces inflammation potentially by increasing the population of *Bifidobacterium* [113]. However, the mechanism for this and the effects of different fibre types on the microbiome remain unclear due to the limitations of these being correlative studies. A double-blind parallel trial conducted on 50 overweight 45–70-year-old men found that a 12-week refined fibre diet increased liver fat and may contribute to MASLD development, while a wholegrain diet was not associated with increased liver fat [114].

Prebiotic ingredients, including some types of dietary fibre, are indigestible by human digestive enzymes and serve as substrates to specific bacteria and have shown promise in liver disease. Several prebiotic sources exist, including oats, unrefined wheat, unrefined barely, yacon, pulses as well as indigestible carbohydrates and oligosaccharides. Inulin, a fibre, showed beneficial effects on ALD in a study conducted on mice. In this study, mice were divided into four groups: a control group not fed inulin, another fed inulin and alcohol-fed groups with and without inulin. This study demonstrated that alcohol-fed mice supplemented with inulin had a significantly increased content of SCFA, including butyrate, propionate and valerate, increased M2 macrophages, arginase-1 and interleukin 10 compared to the alcohol-fed group that was not treated with inulin. Further, inulin feeding significantly reduced M1 macrophages, inducible nitric oxide synthase and TNF-α [115]. As a result, inulin supplementation may provide benefits by increasing SCFA synthesis in liver disease. Further research is required to validate other prebiotic ingredients in the context of liver disease.

While dietary fibre and prebiotics have positive effects on the microbiome, simple sugars can have harmful effects mediated by alterations to the intestinal barrier. Fructose, one of the three simple sugars, is primarily metabolised in the liver by the enzyme fructokinase C [116,117]. This enzyme is also significantly expressed in the small intestine. Ishimoto et al. found that when fructose is metabolised in the small intestine of mice, there is a disruption to tight junctions that are not observed in fructokinase knockout mice. It is postulated that this is the reason for increased gut permeability secondary to fructose consumption [118]. This increased gut permeability leads to endotoxins leaking into the portal vein, which in turn contributes to hepatic inflammation. It is important to note that in that study, the mice were fed a 30–45% fructose diet, whereas the typical Western diet would contain 10–15%. Given these effects, the reduction of simple sugars may be a dietary intervention for individuals with cirrhosis and dysbiosis.

High-fat diets have also been shown to alter the microbiome. David et al. showed that a short-term, high-fat diet compared to a high-fibre plant-based diet, increased the relative abundance of bile-tolerant bacteria [119]. While this trial included only nine subjects, a larger study (*n* = 217) by Wan et al. showed that a higher fat diet increased the relative abundance of *Alistipes* and *Bacteroides* and decreased the relative abundance of *Faecalibacterium* [120]. Additionally, a recent meta-analysis concluded that high-fat diets resulted in dysbiosis, promoting inflammation [121]. High-fat diets are, by definition, low in fruit, wholegrain and legumes and, therefore, low in dietary fibre, which probably explains the effect on the gut microbiota. In any case, given the minimal data collected (only 29 human faecal samples), the differences between experimental food sources and complex human diets as well as the inter- and intra- individual variation in the microbiome, larger studies should be conducted for these results to be conclusive. It is also important to note that there are different types of fat in the diet, and although high fat intake is associated with negative changes, not all fats are harmful to the microbiome. One study has shown that intake of omega-3 fat correlated with higher microbiota diversity and positive changes to microbiome composition in 876 twins [122].

### 7.2. Targeting the Microbiome Directly

Antibiotics can be used to target the microbiome. The use of nonabsorbable antibiotics to target the microbiome in liver disease is attractive to reduce broader effects. Rifaximin is one such antibiotic with a wide range of antimicrobial activity and minimal drug interactions and has shown promising results in patients with cirrhosis. A placebo-controlled, double-blind study conducted on 38 patients with cirrhosis and hepatic encephalopathy found that rifaximin-α led to a resolution of overt and covert hepatic encephalopathy, reduced infection likelihood, decreased oralisation of the gut and lower systemic inflammation. Further, this study found that rifaximin-α largely mediates its anti-inflammatory effects via gut barrier repair, in turn reducing bacterial translocation and endotoxemia [123]. However, despite the promise that antibiotics show as a viable therapy that could be implemented relatively soon, concerns surrounding drug resistance and their lack of specificity remain. As a result, other bactericidal therapies that can be engineered to be more specific may be more desirable in eliminating dysbiosis. 

Bacteriophages are viruses that usually target only a single bacterial species which can undergo lytic or lysogenic replication after infecting a host bacterium. In lytic replication, the host cell resources are rapidly utilised to create viral genomes and capsid proteins, causing the cell to die and more copies of the virus to be released [124]. This may prove to be a useful method of killing targeted bacterial species that are resistant to antibiotics because the continual evolution of these viruses overcomes bacterial defences via an independent mechanism of action from antibiotics. One study conducted on 26 participants without alcohol use disorder, 44 with alcohol use disorder and 88 with alcoholic hepatitis found that bacteriophages were able to specifically target *Enterococcus faecalis*, a bacterium that is associated with more severe liver disease [125]. As a result, these bacteriophages, if developed to target pathobionts in dysbiosis, may provide a new bactericidal tool that may be deployed to improve microbiome health and hence hepatic fibrosis.

Probiotics are live bacteria or yeasts, which, when consumed, may provide benefits by supporting a healthy microbiome. A study of 58 patients with MASLD administration of a multiprobiotic over 8 weeks was able to improve fatty liver, aminotransferase activity and TNF-α and interleukin-6 levels [126]. Further, a phase I trial conducted on patients with cirrhosis demonstrated that the administration of *Lactobacillus rhamnosus GG* was safe and was able to modulate the microbiome, decreasing *Enterobacteriaceae*, endotoxemia and TNF-a while increasing *Lachnospiraceae* [127]. Further, in mice with induced liver fibrosis, *Lactobacillus rhamnosus GG* supplementation was associated with a reduction in liver enzymes. In addition, *Saccharomyces cerevisiae* and *Lactobacillus acidophilus* protected mice from inflammation and hepatic oxidative stress by altering signalling pathways, further supporting the efficacy of probiotics to modulate the microbiome and improve health [128]. A meta-analysis published in 2016, analysing results from 14 studies including 1152 individuals with cirrhosis, found that probiotics were effective in improving minimal hepatic encephalopathy and preventing its progression [129]. However, the same has not been found in primary sclerosing cholangitis, where a small pilot trial found no significant changes to liver markers after three months [130]. One method of overcoming this may be to engineer probiotics in order to target or improve specific effects more effectively; this is currently being explored in clinical trials [131]. Additionally, it is important to note that dietary supplements of probiotics are often ineffective due to strain and dosing differences between studies and due to personalised responses to probiotics [132]. Further, many of these products were developed prior to a well-established understanding of the composition of the microbiome. As such, many of the commercially available probiotic formulations are ineffective at truly altering the microbiome.

FMT is another treatment modality that may become useful in treating liver disease. FMT can be administered via different routes and is mainly used to treat persistent infections with *Clostridioides difficile.* However, our understanding of the ecological forces that affect the microbiome is still limited, so continued research examining mechanisms, safety, efficacy and methods of collection and transplantation is vital for safer and more effective FMT [133]. A randomised control study of 47 MASLD patients who received FMT and 28 MASLD patients who instead received probiotics orally found that FMT was able to decrease liver steatosis and improve dysbiosis, in turn attenuating MASLD [134]. Unfortunately, the current body of FMT research in the context of liver disease is limited, often with small sample sizes and without conclusive evidence of long-term efficacy due to the logistical and regulatory difficulties in conducting these trials. If FMT is to become a widespread and accepted practice, high-quality evidence identifying optimal, safe protocols and indications is required. Current research concludes that FMT had no increase in adverse event rates in severe alcoholic hepatitis, MASH and cirrhosis [135,136,137]. Furthermore, in cirrhosis, FMT via oral capsules after antibiotic use has been shown to be safe in the long term [138]. These findings, however, need to be confirmed in larger cohorts with blind, randomised control trials that demonstrate a clear benefit before clinical adoption can be achieved.

### 7.3. Metabolites and Their Pathways

The interconnection between the microbiome and liver in the bile acid pathway could be leveraged to prevent fibrosis by reducing intestinal permeability or altering the microbiome [94,139,140]. This is possible by treating patients with exogenous FXR agonists. When FXR activation occurs in ileal enterocytes, the expression of fibroblast growth factor 19 (FGF19) is induced. This reaches the liver and, through the portal circulation, activates the fibroblast growth factor receptor 4 (FGFR4)/b-klotho complex on hepatocytes and inhibits the expression of cytochrome P7A1, which reduces bile acid synthesis. This creates a negative feedback loop resulting in reduced FXR activation and is therefore associated with liver fibrosis [68,94,139]. In the ‘Farnesoid X nuclear receptor ligand obeticholic acid for non-cirrhotic, non-alcoholic steatohepatitis’ (FLINT) study, obeticholic acid (OCA), an FXR agonist, was shown to improve primary histological outcomes significantly and reduce liver fibrosis in MASH [140]. Some mechanisms of FXR activation that induce these outcomes include the reduction in inflammatory mediator expression in HSCs via the induction of peroxisome proliferator-activated receptor gamma (PPARγ), inhibition of TGF-beta signalling, sensitising HSCs to apoptotic signals, reducing HSC contractility, mitigating collagen expression and increasing matrix metalloprotease activity [94]. The FLINT study encouraged the exploration of more efficient and safer FXR agonists of which several are undergoing clinical trials with promising results. One such compound, cilofexor, in a double-blind, placebo-controlled phase 2 trial of 140 patients with MASH, was found to be well tolerated and reduced hepatic steatosis and improved liver markers and serum bile acids over a 24-week period [141].

Another means of modulating bacterial byproducts is via carbon nanoparticles such as Yaq-001, a nonabsorbable carbon that can absorb bacterial products such as LPS effectively. As a result, it may be used to counteract gut microbiota alterations and translocation of bacterial-derived products in patients with advanced chronic liver disease. In a study of 12 mice with induced cirrhosis via biliary duct ligation and 12 control mice, increases in Firmicutes and decreases in Bacteroidetes as well as significant reductions in LPS-induced ROS have been demonstrated [142]. Before these treatments are available to patients, their safety and efficacy in humans must be tested; this is ongoing through the CARBALIVE study [143].

## 8. Conclusions

The interactions between the microbiome and the liver are complex and require significantly more research to be fully elucidated. A significant bottleneck to progress has been the difficulty in establishing the mechanistic links between dysbiosis and liver disease. The current literature provides valuable insights into the associations between dysbiosis and liver disease but is incomplete as data on the small intestinal microbiota, which is likely to play an important role in liver disease, is lacking. In metabolic-related liver disease, it was most commonly found that *Ruminococcaceae* and Bacteroidetes decreased in relative abundance, while *Escherichia*, *Lactobacillus*, *Streptococcus*, *Blautia* and *Prevotella* increased in relative abundance. In ALD, Bacteroidetes and Firmicutes had a lower relative abundance, while *Veillonellaceae* was found to have a higher relative abundance. In cirrhosis, decreases in the relative abundance of *Lachnospiraceae* and increases in *Enterobacteriaceae* were common findings. Differences may be explained by environmental and geographical factors, population characteristics, sequencing and diagnostic tools as well as the severity of the disease. Unfortunately, many studies aimed at finding a correlation between the microbiome and liver disease were conducted on small cohorts. With larger cohorts and advancing technologies, our understanding of the mechanisms relating the microbiome to liver disease will improve. Despite the current level of understanding of the interactions between dysbiosis and fibrosis in liver disease, the microbiome offers a novel therapeutic dimension to mitigate liver disease. By gaining a greater understanding of how the microbiome is involved with liver fibrosis, treatments that target the microbiome may provide a promising modality to limit and reverse hepatic fibrosis.

## Figures and Tables

**Figure 1 pathogens-12-01087-f001:**
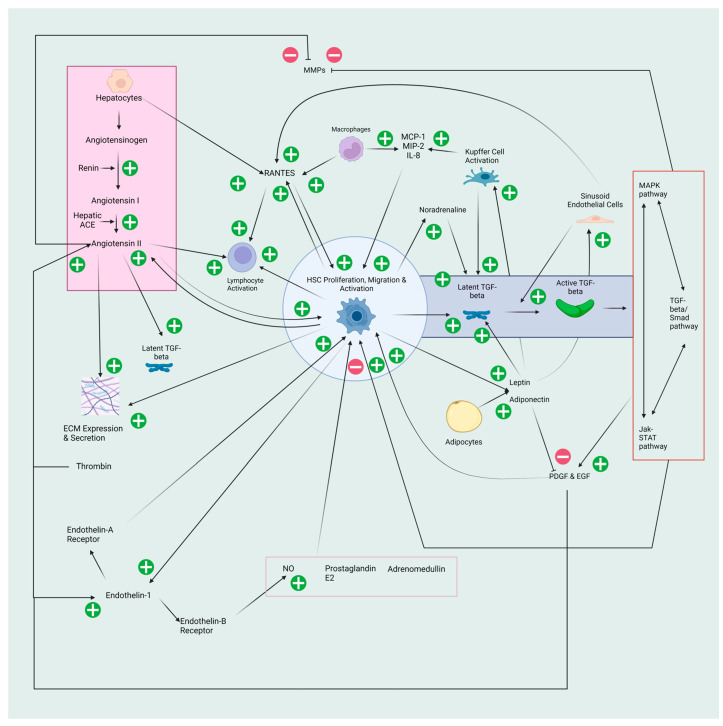
General fibrotic pathways in the liver. Hepatocyte injury leads to hepatocyte apoptosis, the release of reactive oxygen species (ROS) and fibrogenic mediators and capillarisation of sinusoids, which induces the release of latent transforming growth factor-beta (TGF-beta) from hepatic stellate cells (HSCs) and Kupffer cells. This is also stimulated by angiotensin II, part of the renin−angiotensin system (RAS), leptin from adipocytes and HSCs and noradrenaline produced by dopamine beta dehydroxylase found in HSCs. Latent TGF-beta is activated with sinusoid capillarisation and interaction with fibrotic proteins, including collagen IV, fibrinogen and urokinase-type plasminogen activator. TGF-beta activates HSCs via the crosstalk between the TGF-beta/suppressor of mothers against decapentaplegic (SMAD), classical mitogen-activated protein kinase (MAPK) and Janus kinase-signal transduction and activators of transcription (Jak-STAT) pathways. These pathways are also responsible for the inhibition of matrix metalloproteases (MMPs) and increased action of the platelet-derived growth factor (PDGF) and epidermal-derived growth factor (EGF), which both stimulate angiotensin II release and activate HSCs. Hepatocytes store and release angiotensinogen, which is converted to angiotensin I by systemic renin. Angiotensin I is then converted to angiotensin II by hepatic angiotensin-converting enzyme (ACE). Angiotensin II is released by HSCs. This is stimulated by thrombin and endothelin-1. Angiotensin II activates HSCs and lymphocytes and stimulates the expression and secretion of extracellular matrix (ECM) while inhibiting MMPs. Also secreted by hepatocytes as well as sinusoidal epithelial cells, Regulated upon Activation, Normal T Cell Expressed and Presumably Secreted (RANTES), increases HSC and lymphocyte activation. Monocyte chemotactic protein type-1 (MCP-1), macrophage inflammatory protein-2 (MIP-2) and interleukin-8 (IL-8) are released by Kupffer cells and macrophages to increase HSC activation. Adiponectin and leptin are both released from HSCs and adipocytes. Adiponectin inhibits PDGF. Leptin acts on sinusoid epithelial cells and Kupffer cells and increases latent TGF-beta release. Endothelin-1 acts on the endothelin-A receptor to increase HSC activation and acts on the endothelin-B receptor to release nitric oxide (NO), which inhibits HSC activation. Prostaglandin E2 and adrenomedullin also inhibit HSC activation. (Created with BioRender.com; accessed on 21 July 2023).

**Figure 2 pathogens-12-01087-f002:**
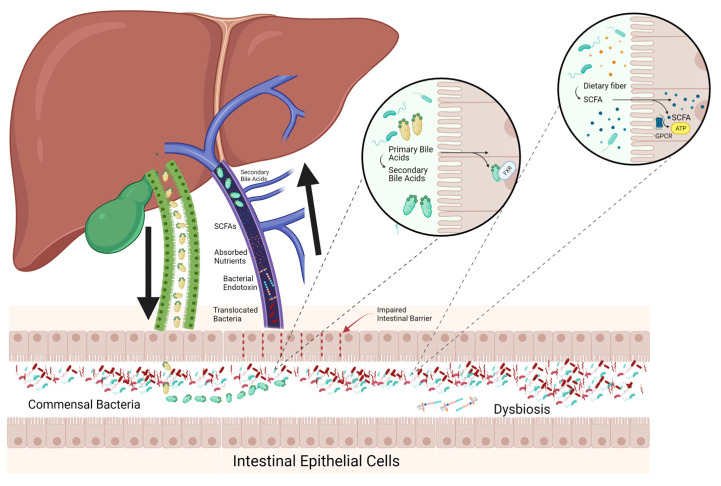
Bidirectional connection between the liver and gut. Primary bile acids produced by the liver are transported to the gut via biliary circulation. In the gut, bacteria convert primary bile acids to secondary and tertiary bile acids that act on farnesoid X receptors (FXR). The portal circulation connects the gut to the liver and is involved in the transport of nutrients, secondary bile acids, short-chain fatty acids (SCFAs), bacterial endotoxins and translocated bacteria to the liver from the gut. SCFAs are produced when prebiotic ingredients, including dietary fibre, are processed by the microbiota and act on G-protein-coupled receptors (GPCR) to produce adenosine triphosphate (ATP) in enterocytes. In liver disease, dysbiosis can occur, which will impair the intestinal barrier and increase the translocation of bacteria and endotoxins to the liver, which can cause injury. (Created with BioRender.com; accessed on 21 July 2023).

**Table 1 pathogens-12-01087-t001:** Studies outlining associations between dysbiosis and metabolic- and alcohol- related liver disease.

Study	Year	Design	Sample Size	Findings
Metabolic Dysfunction-Associated Steatotic Liver Disease—Animal Studies				
Le Roy et al. [70]	2013	Animal Study		Germ-free mice exhibited the same glycaemia profile as their donor post FMT. The hyperglycaemic mice developed hepatic macrovesicular steatosis
Metabolic Dysfunction-Associated Steatotic Liver Disease—Human Studies				
Jiang et al. [71]	2015	Cross-Sectional Study	MASLD: *n* = 53 Healthy individuals: *n* = 32	Decreased *Alistipes*, *Prevotella* and *Ruminococcaceae*; Increased *Escherichia*, *Anaerobacter*, *Lactobacillus* and *Streptococcus*
Raman et al. [72]	2013	Observational Case-Control Study	MASLD: *n* = 30 Healthy individuals: *n* = 30	Decreased *Ruminococcaceae*; Increased *Lactobacillus* and select Firmicutes
Zhu et al. [73]	2012	Case-Control Study	MASH: *n* = 22 Obesity: *n* = 25 Healthy individuals: *n* = 16	Decreased Firmicutes and Actinobacteria; Increased *Bacteroides* and Proteobacteria
Meijnikman et al. [74]	2020	Prospective Study and Intervention Study	Prospective: *n* = 146 Intervention arm: MASLD: *n* = 10 Obesity: *n* = 10	Blood ethanol was higher in MASLD and increased with disease progression to MASH; Inhibition of alcohol dehydrogenase led to a 15× increase in peripheral blood ethanol concentrations in MASLD but disappeared after antibiotic treatment
Shen et al. [75]	2017	Cross-Sectional Study	MASLD: *n* = 35 Healthy Controls: *n* = 22	Lower gut microbiota diversity; Increased Proteobacteria, Fusobacteria, *Lachnospiraceae*, *Enterobacteriaceae*, *Erysipelotrichaceae*, *Streptococcaceae*, *Escherichia* and *Blautia*; Decreased Bacteroidetes, *Prevotellaceae* and *Ruminococcaceae*
Del Chierico et al. [76]	2016	Case-Control Study	MASLD: *n* = 27 MASH: *n* = 26 Obesity: *n* = 8 Controls: *n* = 54	Decrease of *Oscillospira*; Increases of *Ruminococcus*, *Blautia*, and *Dorea*
Mouzaki et al. [77]	2013	Cross-Sectional Study	MASLD: *n* = 11 MASH: *n* = 22 Healthy Controls: *n* = 17	Decreased Bacteroidetes; Increased *Clostridium coccoides*
Michail et al. [78]	2015	Cross-Sectional Study	MASLD: *n* = 13 Obesity: *n* = 11 Healthy Controls: *n* = 26	Increased *Gammaproteobacteria* and *Prevotella* and significantly higher levels of ethanol
Wong et al. [79]	2013	Longitudinal Study	MASH: *n*= 16 Healthy Controls: *n*= 22	Decreased *Faecalibacterium* and *Anaerosporobacter*; Increased *Parabacteroides* and *Allisonella*
Schwimmer et al. [80]	2019	Cross-Sectional Study	MASLD: *n* = 87 Obesity: *n*= 37	High relative abundance of *Prevotella copri* was associated with more severe fibrosis
Iwaki et al. [81]	2021	Cross-Sectional Study	Obese MASLD: *n* = 51 Nonobese MASLD: *n* = 51 Control: *n* = 87	Decreased *Eubacterium* in non-obese vs. obese MASLD
Yun et al. [82]	2019	Cross-Sectional Study	Lean MASLD: *n* = 27 Obese MASLD: *n* = 49 Control: *n* = 192	Decrease in *Desulfovibrionaceae* for lean vs. obese MASLD
Alcohol-related Liver Disease—Animal Studies				
Bull-Otterson et al. [83]	2013	Animal Study		Decreased Bacteroidetes and Firmicutes; Increased *Alcaligenes* and *Corynebacterium*
Yan et al. [84]	2011	Animal Study		Increased Bacteroidetes and Verrucomicrobia; Downregulation of bactericidal c-type lectins Reg3b and Reg3g protein expression in the small intestine
Llopis et al. [85]	2015	Animal Study		Decreased *Faecalibacterium*; Increased *Bacteroides*, *Bilophila*, *Alistipes*, *Butyricimonas* and *Clostridium* cluster XIVa
Martino et al. [86]	2022	Animal Study		No evidence of microbial metabolism of ethanol; Microbiota activate acetate dissimilation when exposed to ethanol
Alcohol-related Liver Disease—Human Studies				
Mutlu et al. [87]	2012	Cross-Sectional Study	ALD: *n* = 19 Alcohol use disorder without ALD: *n* = 28 Healthy individuals: *n* = 18	Decreased Bacteroidetes; Increased Proteobacteria was persistent after extended periods of sobriety. This was correlated with high levels of serum endotoxin and decreased connectivity of the microbial network unique to the patients that with ALD
Smirnova et al. [88]	2020	Cross-Sectional Study	Alcoholic Hepatitis: *n* = 34 Alcohol Use Disorder: *n* = 20 Healthy Controls: *n* = 24	Increased *Atopobium*, *Fusobacterium* and several genera of *Veillonellaceae*; Decreased *Lachnospiraceae* and *Ruminococcaceae*; Microbial taxa did not distinguish between disease severity
Lang et al. [89]	2020	Multicentre Observational Study	Alcoholic Hepatitis: *n* = 75 Alcohol Use Disorder: *n* = 43 Healthy Controls: *n* = 14,	Severe alcoholic hepatitis had decreased *Akkermansia* and increased *Veillonella* compared to less severe disease
Cirrhosis—Human Studies				
Bajaj et al. [90]	2014	Cross-Sectional Study	Cirrhosis: *n* = 219 - Compensated: *n* = 121 - Decompensated: *n* = 98 Healthy individuals: *n* = 25	Decreased Clostridiales XIV, *Ruminococcaceae*, *Lachnospiraceae*, *Veillonellaceae*, and *Porphyromonadaceae*; Increased *Enterococcaceae*, *Staphylococcaceae*, and *Enterobacteriaceae*
Chen et al. [91]	2011	Cross-Sectional Study	Cirrhosis: *n* = 36 Healthy individuals: *n* = 24	Decreased Bacteroidetes and *Lachnospiraceae*; Increased Fusobacteria, *Enterobacteriaceae*, *Veillonellaceae* and *Streptococcaceae*
Qin et al. [92]	2014	Cross-Sectional Study	Cirrhosis: *n* = 98 Healthy individuals: *n* = 83	28 bacterial species increased in relative abundance; 38 bacterial species decreased; 54% of the increased species were of buccal origin
Bajaj et al. [93]	2018	Cross-Sectional Study	Turkish cirrhosis: *n* = 93 Turkish healthy individuals: *n* = 46 USA cirrhosis: *n* = 109 USA healthy individuals: *n* = 48	Turkish cohort had greater microbial diversity than the American participants; Turkish patients showed no change in diversity between individuals with cirrhosis and controls

## Data Availability

Not applicable.

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
