# Peer review of "The Role of the Gastrointestinal Microbiome in Liver Disease"

_pathogens, 2023, doi:10.3390/pathogens12091087_

Round 1

Reviewer 1 Report

This is an excellent review paper on an emerging topic of dysbiosis in patients with chronic liver disease. I do not have any major criticism as the paper is very clear, detailed and well organized. 

My 2 minor points are the following:

1. Introduction- line 43: infections in patients with cirrhosis should be listed in addition to others as it is evidence that patients with cirrhosis are immunocompromised. 

2. patients with lean fatty liver have different microbiome than obese patients with MAFLD and authors might want to elaborate more on this. 

Once again, this is very well written paper, informative and contributes to the field of hepatology. I have enjoyed reading it and would like to congratulate authors on a job well done!

Author Response

Thank you all for your helpful feedback and for taking the time to read over our work. We greatly appreciate your opinions and believe it has greatly helped to make this manuscript better.

In response to Reviewer 1

Thank you for taking the time to review our article and for your feedback.

My 2 minor points are the following:

  1. Introduction- line 43: infections in patients with cirrhosis should be listed in addition to others as it is evidence that patients with cirrhosis are immunocompromised. 

We agree and have made the change as seen on line 47.

  1. patients with lean fatty liver have different microbiome than obese patients with MAFLD and authors might want to elaborate more on this. 

We agree and have added some information about studies that demonstrate a difference in the microbiome in lean vs obese patients with MASLD as seen on lines 336-343.

Reviewer 2 Report

This manuscript addresses a timely topic and gives a broad overview of the actual knowledge of the gastrointestinal microbiome and its influence on liver fibrosis. However, some revisions are needed:

1.      Spelling, punctuation and grammar must be revised throughout the entire manuscript.

2.      The manuscript rather summarizes liver inflammation in NASH and ALD, while the exact mechanisms of HSC activation and fibrosis generation are too broadly mentioned (ref. to my comments for chapter 4, 5 and 6). Either the paper is revised substantially, or the authors add another chapter on how chapters 4-6 interrelate with fibrogenesis, or the title of the manuscript should be adjusted and the term liver fibrosis changed to liver disease.

3.      The abbreviations need to be checked and revised:

o    Before using an abbreviation, introduce it. When you have introduced the abbreviation, please only use this consistently.

Some examples:

o    Line 56: ROS (abbreviation wasn’t introduced)

o    Line 66: TGF-beta (abbreviation wasn’t introduced)

o    Line 83: TGF-β (beta or β?)

o    Line 118: HLA (abbreviation wasn’t introduced)

o    Line 122: primary biliary cholangitis (PBC) and primary sclerosing cholangitis à There is also an abbreviation: PSC

o    144: GIT (abbreviation wasn’t introduced)

o    533: FLINT study (abbreviation wasn’t introduced)

4.     Some citations (numbers) need to be checked (f.ex. line 183)

5.     The Bacteria names in the text are non-cursive but in the table they are (consistency between text and table)

6.     There is an incorrect punctuation (line 17)

7.     Line 17: The interaction between the liver and the microbiome à Which microbiome? Hepatic? Gut? Oral?

8.     The entire passage from Line 66 onwards seems like a random composition of inflammatory mediators: Why did the authors choose these mediators and pathways? Either refer to the selected mediators/pathways later in the review or give more details for the sepected ones:

o    Line 66f: TGFb activated by… à Where does TGFb come from? Please add, which cells secrete it.

o    Line 73: Release of inflammatory mediators à which? Please add.

o    Lines 74-79: “Other vasoconstrictors (including Endothelin-1 and noradrenaline) increase fibrosis while vasodilators (including nitric oxide and relaxin) exert antifibrogenic effects.” Please give more details about how these mediators are secreted, from which cells, etc.

9.     Chapter 2.3: Which virus? HIV? HCV? HBV? Please specifiy.

10.  Figure 1: Revise figure and figure legend:

o    add MCP-1 and CCL5 etc mentioned in the figure.

o    What is the green cell (sinus? Bacteria? HSC?) in the upper right corner?

o    What are the blue dots above the liver?

11.  Figure 2:

o    ATP à It is in the figure but not explained in the legend. Please introduce the abbreviation; you may add it in the context of GPCR in figure legend

o    Please add GPCR in the Figure

12.   Line 126: what do you mean by “liver nodules”? Please use another term.

13.   Line 129: Which inflammatory cytokines?

14.   Line 237: NASH refers to “non-alcoholic steatohepatitis”. Please adjust.

15.   Chapter 4: This is nice to read, but can be shortened or even erased, as it does not become clear, what the interrelation is with this summary and the development of fibrosis. The authors can just refer to their table 1 for an overview over selected studies (nice work!) on microbiome changes in NASH and ALD.

16.   Chapter 5: Also in this chapter does not really draw a line to liver fibrosis. It just starts at about line 379, where activation of HSCs is mentioned. Is  there anything else, that can be added? Some inflammatory pathways in HSCs that are activated by SCFA? Are bacteria found to generate collagene?

17.   Chapters 6.1 and 6.2.: They also fail the scope of the review. Both chapters basically deal with liver inflammation and gut barrier. Nothing is said about an interrelation with HSC activation.

18.   Chapter 6.3: the section about FGF19 is very relevant. More details in how this relates to fibrogenesis should be added.

see above

Author Response

Thank you all for your helpful feedback and for taking the time to read over our work. We greatly appreciate your opinions and believe it has greatly helped to make this manuscript better.

In response to Reviewer 2

Thank you for taking the time to review our article and for your feedback.

  1. Spelling, punctuation and grammar must be revised throughout the entire manuscript.

We have edited the spelling, punctuation and grammar more thoroughly.

  1. The manuscript rather summarizes liver inflammation in NASH and ALD, while the exact mechanisms of HSC activation and fibrosis generation are too broadly mentioned (ref. to my comments for chapter 4, 5 and 6). Either the paper is revised substantially, or the authors add another chapter on how chapters 4-6 interrelate with fibrogenesis, or the title of the manuscript should be adjusted and the term liver fibrosis changed to liver disease.

Thank you for the feedback. Upon review we agree that the link between fibrosis and the content in chapters 4-6 is not strong enough and so have changed the title and adjusted each of these paragraphs to refer to liver disease rather than fibrosis. Additionally we have added an additional chapter 6 which draws a more concrete link between dysbiosis, intestinal permeability and endotoxin production.

  1. The abbreviations need to be checked and revised:

o    Before using an abbreviation, introduce it. When you have introduced the abbreviation, please only use this consistently.

Some examples:

o    Line 56: ROS (abbreviation wasn’t introduced)

o    Line 66: TGF-beta (abbreviation wasn’t introduced)

o    Line 83: TGF-β (beta or β?)

o    Line 118: HLA (abbreviation wasn’t introduced)

o    Line 122: primary biliary cholangitis (PBC) and primary sclerosing cholangitis à There is also an abbreviation: PSC

o    144: GIT (abbreviation wasn’t introduced)

  • 533: FLINT study (abbreviation wasn’t introduced)

We have edited to make sure these instances and any others where abbreviations were not introduced have been resolved.

  1. Some citations (numbers) need to be checked (f.ex. line 183)

We have edited the citations to ensure these errors have been fixed.

  1. The Bacteria names in the text are non-cursive but in the table they are (consistency between text and table)

This has been edited throughout the text to ensure consistency.

  1. There is an incorrect punctuation (line 17)

This has now been resolved. See line 17.

  1. Line 17: The interaction between the liver and the microbiome à Which microbiome? Hepatic? Gut? Oral?

We have changed it to “gastrointestinal microbiome”. See line 18.

  1. The entire passage from Line 66 onwards seems like a random composition of inflammatory mediators: Why did the authors choose these mediators and pathways? Either refer to the selected mediators/pathways later in the review or give more details for the sepected ones:

o    Line 66f: TGFb activated by… à Where does TGFb come from? Please add, which cells secrete it.

o    Line 73: Release of inflammatory mediators à which? Please add.

  • Lines 74-79: “Other vasoconstrictors (including Endothelin-1 and noradrenaline) increase fibrosis while vasodilators (including nitric oxide and relaxin) exert antifibrogenic effects.” Please give more details about how these mediators are secreted, from which cells, etc.

We have revised this section extensively to provide more context regarding the pathways and cells that these mediators are involved in however have tried to do so within the lens of liver fibrosis and so this may not be an extensive discussion of the other roles these mediators play throughout the body.

  1. Chapter 2.3: Which virus? HIV? HCV? HBV? Please specifiy.

       We have added “(such as caused by hepatitis B, C or Delta)” to make this clearer. See line 183.

  1. Figure 1: Revise figure and figure legend:

o    add MCP-1 and CCL5 etc mentioned in the figure.

o    What is the green cell (sinus? Bacteria? HSC?) in the upper right corner?

o    What are the blue dots above the liver?

      This figure has been greatly revised to reflect the changes made to Chapter 2.1 and the legend has also been updated.

  1. Figure 2:

o    ATP à It is in the figure but not explained in the legend. Please introduce the abbreviation; you may add it in the context of GPCR in figure legend

  • Please add GPCR in the Figure

These changes have been made in the figure and we have added “…G-protein-coupled receptors (GPCR) to produce adenosine triphosphate (ATP) in enterocytes.” to the legend on lines 245-246.

  1. Line 126: what do you mean by “liver nodules”? Please use another term.

       We have changed it to “nodules of regenerating parenchyma”. See line 198.

  1. Line 129: Which inflammatory cytokines?

       We have changed this to “due to the release of TGF-beta…”. See line 200.

  1. Line 237: NASH refers to “non-alcoholic steatohepatitis”. Please adjust.

       This has been changed to MASH to reflect the changed nomenclature. See line 162.

  1. Chapter 4: This is nice to read, but can be shortened or even erased, as it does not become clear, what the interrelation is with this summary and the development of fibrosis. The authors can just refer to their table 1 for an overview over selected studies (nice work!) on microbiome changes in NASH and ALD.

       We believe that discussing the studies from the table in chapter 4 allows us to provide more context about each of the studies and therefore provide more insight to the reader about the state of research in regards to liver disease and dysbiosis and so have elected to keep chapter 4 in the same format. We have edited it and added another section describing studies that demonstrate a difference in the microbiome in lean vs obese patients with MASLD to address feedback from another reviewer. See lines 336-343.

  1. Chapter 5: Also in this chapter does not really draw a line to liver fibrosis. It just starts at about line 379, where activation of HSCs is mentioned. Is  there anything else, that can be added? Some inflammatory pathways in HSCs that are activated by SCFA? Are bacteria found to generate collagene?

       In response to your previous feedback we have altered the scope of the report to reflect liver disease rather than liver fibrosis. Additionally, the end of chapter 5 has been expanded upon and placed into a new chapter, chapter 6 which links dysbiosis, intestinal permeability and endotoxin production.

  1. Chapters 6.1 and 6.2.: They also fail the scope of the review. Both chapters basically deal with liver inflammation and gut barrier. Nothing is said about an interrelation with HSC activation.

       Due to the change in the scope of the article form liver fibrosis to liver disease and with the addition of chapter 6 we believe chapter 7.1 and 7.2 are now more appropriate.

  1. Chapter 6.3: the section about FGF19 is very relevant. More details in how this relates to fibrogenesis should be added.

Chapter 7.3 has been revised to more explicitly discuss the mechanisms of FXR activation which lead to the expression of FGF19. See lines 628-637.

Reviewer 3 Report

I read with interest the review of Shalaby et al regarding the link between liver fibrosis and gut microbiome. It is a very well-written review that perfectly addresses the evidence of the possible role of the gut microbiota in the progression of liver diseases. I believe that there are a few issues that need to be addressed.

-          - Suggest you make slight modifications to the first two sections of the review. Include an initial introduction section that briefly explains the relevance of liver fibrosis and its relationship with the gut microbiota, leading to the subsequent sections of the review. The second section should be on fibrosis, including the explanation of its pathophysiology.

-          - Suggest you change the term MAFLD to NAFLD. Recently, the nomenclature has been modified to MASLD (metabolic dysfunction-associated steatotic liver disease), but all the studies mentioned in this review use the term NAFLD.

-          - P4, line 105: The postulated model is the multiple parallel hits hypothesis, not a two-hit model, where many hits derived from the gut and/or the adipose tissue may act in parallel promoting the development of liver inflammation and fibrosis.

-          - P5, line 144: The acronym GIT is not defined.

-     - P7, line 236: Suggest you mention the limitations of the analysis of 16S rRNA sequencing for the study of the intestinal microbiota.

-              - P15, line 510: Suggest you delete “after antibiotic use”.

-     - P15, line 527-539: Have modifications in intestinal microbiota and/or intestinal permeability been demonstrated with FXR agonists? If not, please modify this paragraph.

Author Response

Thank you all for your helpful feedback and for taking the time to read over our work. We greatly appreciate your opinions and believe it has greatly helped to make this manuscript better.

In response to Reviewer 3

Thank you for taking the time to review our research and provide your feedback.

I read with interest the review of Shalaby et al regarding the link between liver fibrosis and gut microbiome. It is a very well-written review that perfectly addresses the evidence of the possible role of the gut microbiota in the progression of liver diseases. I believe that there are a few issues that need to be addressed.

-          - Suggest you make slight modifications to the first two sections of the review. Include an initial introduction section that briefly explains the relevance of liver fibrosis and its relationship with the gut microbiota, leading to the subsequent sections of the review. The second section should be on fibrosis, including the explanation of its pathophysiology.

      We have altered the first section by adding some more signposting to explain the relevance of the microbiome and its relationship in liver disease. See lines 31-34.

-          - Suggest you change the term MAFLD to NAFLD. Recently, the nomenclature has been modified to MASLD (metabolic dysfunction-associated steatotic liver disease), but all the studies mentioned in this review use the term NAFLD.

      We have opted to use the most up to date terminology and instead explain the changes in nomenclature in chapter 2.2. See lines 158-165.

-          - P4, line 105: The postulated model is the multiple parallel hits hypothesis, not a two-hit model, where many hits derived from the gut and/or the adipose tissue may act in parallel promoting the development of liver inflammation and fibrosis.

      We have altered this paragraph describing the multiple parallel hits model. See lines 174-180.

-          - P5, line 144: The acronym GIT is not defined.

      We have fixed this. See line 231.

-     - P7, line 236: Suggest you mention the limitations of the analysis of 16S rRNA sequencing for the study of the intestinal microbiota.

      We have added some information about the limitations of 16S rRNA sequencing in lines 314-317.

-              - P15, line 510: Suggest you delete “after antibiotic use”.

This has been changed. See line 604.

-     - P15, line 527-539: Have modifications in intestinal microbiota and/or intestinal permeability been demonstrated with FXR agonists? If not, please modify this paragraph.

This has been shown in three studies see which have been added as citations. See references 138-140.